# Down Syndrome in FPIES: An Overwhelming and Unexpected Prevalence

**DOI:** 10.3390/jcm11144047

**Published:** 2022-07-13

**Authors:** Valentina Pecora, Maurizio Mennini, Rocco Valluzzi, Vincenzo Fierro, Alberto Villani, Diletta Valentini, Alessandro Fiocchi

**Affiliations:** 1Translational Research in Pediatric Specialities Area, Division of Allergy, Bambino Gesù Children’s Hospital, Istituto di Ricovero e Cura a Carattere Scientifico, Piazza Sant’Onofrio, 4, 00165 Rome, Italy; maurizio.mennini@opbg.net (M.M.); roccoluigi.valluzzi@opbg.net (R.V.); vincenzo.fierro@opbg.net (V.F.); agiovanni.fiocchi@opbg.net (A.F.); 2Pediatric Unit, Pediatric Emergency Department (DEA), Bambino Gesù Children’s Hospital, Istituto di Ricovero e Cura a Carattere Scientifico, Piazza Sant’Onofrio, 4, 00165 Rome, Italy; alberto.villani@opbg.net (A.V.); diletta.valentini@opbg.net (D.V.)

**Keywords:** cow’s milk proteins, Down syndrome, food protein-induced enterocolitis syndrome

## Abstract

Down syndrome (DS) is one of the most common chromosomal anomalies. Gastrointestinal disorders in DS are predominantly related to anatomical anomalies and celiac disease. In 2015, the first two cases of non-IgE-mediated food allergy in patients with DS were described. However, gastrointestinal symptoms experienced by subjects with DS have never been related to a possible non-IgE-mediated food allergy and a Food Protein-induced Enterocolitis syndrome (FPIES). A retrospective descriptive single-center study was conducted. Subjects included were children with acute FPIES who entered our institutional follow-up protocol between January 2013 and January 2020. Among the 85 patients (forty-nine boys—57.6%), ten (11.76%) were children with DS. In our population, the FPIES triggers included different foods (such as milk, egg, fruit, fish, wheat, soy, beef, etc.). Nine patients with DS showed FPIES reactions after ingesting cow’s milk (one even with beef and three with soy), while the last one was affected by FPIES to fish. Considering the subgroup of patients affected by cow’s milk FPIES (40 subjects overall), 22.5% had a diagnosis of DS. Patients with DS experienced acute FPIES reactions with a severity degree slightly higher than that reported in other patients, ranging from mild-moderate to severe or very severe. During the acute reactions, the patients with DS showed increased white blood cell production, absolute neutrophil count and C-reactive protein levels. This series provides a starting point for novel hypothesis-testing clinical research and possible specific immunological alterations in FPIES children with or without DS.

## 1. Introduction

Down syndrome (DS) is the most prevalent survivable chromosomal disorder, which is attributed to the duplication of chromosome 21 and the subsequent alteration of the dosage of genes located on this chromosome. Individuals with DS are at increased risk of death, both overall and specifically from congenital heart defects, dementia and leukemia, compared to normal individuals [1]. They are also at high risk for gastrointestinal (GI) disorders. Congenital GI anomalies are present in about 6.7% of individuals with DS: esophageal atresia/tracheoesophageal fistula (0.4%), pyloric stenosis (0.3%), duodenal stenosis/atresia (3.9%), Hirschsprung’s disease (0.8%), and anal stenosis/atresia (1.0%) [2]. Furthermore, DS is associated with an increased risk for celiac disease (CD), an immune-mediated GI disorder characterized by inflammation of the small intestine on exposure to gluten, a protein found in wheat, barley and rye.

An association with multiple immune aberrations is well established [3,4]. Compared with a healthy population, patients with DS are significantly more susceptible to infectious diseases and have a higher frequency of malignant neoplastic diseases (especially leukemia) in addition to the enhanced predisposition to autoimmune diseases. Several studies indicate that DS patients show low IgE-specific allergic sensitization rates [5,6].

Four-cytokine receptor genes are located on human chromosome 21q22.11, encoding the α and β subunits of the interferon-α receptor, the β subunit of the IL-10 receptor, and the second subunit of the interferon-γ receptor [7]. Overexpression of chromosome 21-gene products causes dysregulation of the pro-inflammatory cytokines characterized by an increase of TNF-α and IFN-γ levels with a decrease in plasma of IL-10 concentration [8]. These pro-inflammatory cytokines might be useful as early biomarkers of the disorders associated with DS [9].

The first report of non-IgE-mediated gastrointestinal food allergy in DS dates to 2015 [10]. In a letter to the editor, Wakiguchi and colleagues describe the cases of two Japanese patients with DS and their clinical history of severe and long-lasting gastrointestinal symptoms associated with cow’s milk exposure. Nevertheless, a possible association between DS and food allergy was never suggested.

## 2. Materials and Methods

We retrospectively reviewed cases of acute FPIES included in our institutional follow-up protocol between January 2013 and October 2020. The clinical diagnostic criteria applied to establish the diagnosis of FPIES agreed with the first international consensus statement [11]. In our population, the offending foods responsible for FPIES acute reactions included different foods, such as animal-source foods, seafood, and plants (wheat, rice, fruit and vegetables).

The food reactions were classified into mild-moderate, severe to very severe according to the FPIES guidelines [11]. In the mild-moderate reactions, the subjects presented less than three episodes of emesis, pallor and/or decreased activity level and rarely diarrhea. In the severe reactions, more than three episodes of emesis, pallor, lethargy, and dehydration were observed. Finally, the very severe reactions were characterized by more than three episodes of emesis, pallor, lethargy, dehydration, cyanotic appearance and water/bloody diarrhea.

Within the diagnostic procedure, all subjects underwent a complete allergy assessment. Skin prick tests (SPTs) for the triggering food allergens were performed by the standard technique [12]. Serum-specific IgE to the trigger food allergen (s) was determined using a fluorescent enzyme immunoassay (ImmunoCAP System—Thermo Scientific, Uppsala, Sweden) according to the manufacturer’s instructions with a positive cutoff point set at 0.1 kUA/L.

If all the necessary diagnostic criteria were not met, the subjects underwent an oral food challenge to confirm the FPIES diagnosis. The challenge procedure was performed according to international guidelines [13], and intravenous access was always secured. Fresh foods were administered at a dose of 0.3 g of the food protein per kilogram of body weight in three equal doses over 30 min, generally not to exceed a total of 3 g of protein or 10 g of total food (100 mL of liquid for milk) for an initial feeding. Only reactions to OFC occurring within 6 h are reported. Written informed consent was obtained from all procedures performed.

## 3. Results

Among the 85 FPIES patients (forty-nine boys—57.6%), ten (11.76%; with a male-to-female ratio of 1.5) were affected by DS. The clinical characteristics of the 85 FPIES patients (subjects with DS compared with non-DS individuals) and the type of reactions are reported in Table 1.

Of all deliveries, 43 (50.6%) were cesarean sections, while 35 (41.2%) and 7 (8.2%) underwent vaginal and dystocic delivery, respectively. The comparison of the ten individuals with DS with the remaining population did not provide a remarkable difference in the type of delivery, even if the subjects with DS showed a lower prevalence of cesarean section.

Though the FPIES triggers included different foods (such as milk, egg, fruit, fish, wheat, etc.), nine patients with DS showed acute FPIES reactions after the ingestion of cow’s milk (one of them even with beef and three with soy), while the last one was affected by FPIES to fish. Considering the subgroup of patients affected by FPIES from cow’s milk (40 subjects overall), 22.5% showed DS. The comparison within the subgroup of milk FPIES patients with or without DS is reported in Table 2.

Across the population of patients without DS, the mean age of clinical onset is 6.20 months (SD 6.93—median 5—range from 0 to 48 months), while for the group of children with DS, the mean age drops to 2.39 months (SD 3.27—median 0.5—range from 0 to 8 months). Early onset in patients with DS was confirmed, however, within the milk FPIES subgroup [1.65 months (SD 2.22) vs. 3.13 months (SD 4.27)]. Comparing the subjects with DS with the remaining population, the number of episodes before the diagnosis for each patient was higher in patients with DS [4.50 (SD 3.50) vs. 2.82 (SD 1.39)]. When we considered the milk FPIES subgroup, the same trend was observed [4.66 (SD 3.67) vs. 2.38 (SD 1.08)]. Positive SPT and sIgE determination to alfa-lactoalbumin and beta-lactoglobulin were observed in three patients (one in the DS subgroup) without a suggestive clinical history of an IgE-mediated allergy. Patients with DS experienced acute FPIES reactions with a severity degree slightly higher than that reported in other patients, ranging from mild-moderate to severe or very severe. The mean duration of time interval from the ingestion of offending food to the onset of FPIES symptoms was 2.04 h (SD 1—median 2) in the non-DS population and 1.7 h (SD 0.84—median 1.5) in the subgroup with DS. Considering the patients with milk FPIES, the acute reactions started with a shorter time interval in subjects with DS [1.55 (SD 0.76) vs. 2.27 (SD 1.33)]. Of the overall 44 acute FPIES reactions experienced by the ten patients with DS, six were managed in our pediatric emergency unit, whereas in two cases, the onset of symptoms occurred during a follow-up oral challenge. In these six circumstances, blood samples were drawn as previously described [14], showing increases in white blood cell production, absolute neutrophil count and C-reactive protein levels.

## 4. Discussion

To date, there has been no reported connection between patients with DS and a possible non-IgE-mediated food allergy whereby gastrointestinal symptoms were experienced. Since the 1990s, the Down Syndrome Medical Interest Group (DSMIG), a panel of multidisciplinary clinical experts on DS, recommended that children with DS be screened for CD between the ages of two and three [15]. These recommendations were retired in 2011 when the American Academy of Pediatrics published new guidelines recommending serologic screening for CD only in symptomatic children with DS at each preventative care visit, beginning at age the age of one for children on a diet containing gluten [16]. Our data indicate that FPIES should be considered in the differential diagnosis of GI disorders in DS.

FPIES reactions often appear in the first weeks or months of life and usually occur upon introduction of the offending food for the first time. An early onset characterizes subjects affected by cow’s milk or soy FPIES. There is a remarkable uniformity in the characteristics of the acute FPIES reactions represented by (i) the onset of symptoms at least 60 min after the offending food ingestion; (ii) the absence of classic IgE-mediated allergic skin or respiratory symptoms; (iii) the strict avoidance of the suspected food trigger guarantees an absence of new acute reactions.

In FPIES triggered by cow’s milk, a CD4+ T-cell-proliferative response and Th2 cytokines production after casein stimulation were found to be similar to those of control subjects, while after positive oral food challenges (OFCs), higher levels of IL-10 and of IL-8 have been recently reported [17]. In line with these findings, Wakiguchi and colleagues [10] speculated that the impaired IL-10 production described in patients with DS could support the severe and long-lasting symptoms. In addition, an extended activation of cells of the innate immune system after food challenge, including NK cells, has been recently found [18]. Our preliminary data support the hypothesis that abnormalities of the host innate immune response could induce a misrecognition of specific foods in FPIES children [14]. These last considerations could be strengthened by the demonstration of acute FPIES reactions in patients with DS.

This study has all the limitations inherent to a retrospective record review. As our hospital represents a national reference center for DS with about 800 cases in follow-up, our data could overestimate the incidence of DS in FPIES. A questionnaire survey is underway to calculate the incidence of FPIES in DS. However, if the frequency were limited to 10/800, this would significantly exceed the estimated frequency of FPIES in the general population. In any case, this report may provide a starting point for novel hypothesis-testing clinical research and possible specific immunological alterations in DS.

In conclusion, this retrospective record review indicates the necessity to verify the prevalence of DS in other FPIES caseloads. If confirmed, they would indicate a 53-times higher prevalence of DS among children with FPIES than in the general population (11.76%) compared to the general European population (0.22%) might have important implications for the diagnostic suspicion, workup, and treatment by health professionals. They should seriously consider FPIES when children with DS experience GI food reactions. More generally, considering the immunological alterations of DS, our data focuses attention on the pathogenesis of FPIES by strengthening the hypothesis that it is an immunologically complex disease.

## Figures and Tables

**Table 1 jcm-11-04047-t001:** Clinical characteristics of DS vs. non-DS patients with FPIES.

No. of FPIES Patients	85	
	*DS Population 10*	*Non-DS Population 75*
Sex		
Male (%)	6 (60%)	43 (57.3%)
Female (%)	4 (40%)	32 (42.7%)
Route of delivery (%)		
Vaginal delivery	4 (40%)	31 (41.3%)
Dystocic delivery	2 (20%)	5 (6.7%)
Cesarean section	4 (40%)	39 (52%)
Age (months) at first reaction		
Mean (SD)/median	2.36 (2.96)/0.5	6.20 (6.93)/5
Range	0–8	0–48
No. of episodes before diagnosis		
Mean (SD)/median	4.11 (3.48)/3	2.85 (1.39)/3
Range	1–13	1–6
Offending foods, *n* (%)	*44 acute reactions*	*272 acute reactions*
Milk	36 (81.8%)	96 (35.4%)
Rice		57 (20.9%)
Egg		32 (11.8%)
Fish (cod, sole)	3 (6.8%)	31 (11.4%)
Wheat		19 (6.9%)
Fruit (apple, pear, peach, apricot)		16 (5.9%)
Beef	1 (2.4%)	4 (1.5%)
Soy	4 (9%)	2 (0.7%)
Others		15 (5.5%)
Time (h) to onset of symptoms		
Mean (SD)/median	1.7 (0.84)/1.5	2.04 (1)/2
Range	1–4	1–7
Severity of the acute FPIES reactions (%)	*44 acute reactions*	*272 acute reactions*
Mild-moderate	18 (40.9%)	108 (39.7%)
Severe	11 (25%)	102 (37.5%)
Very severe	15 (34.1%)	62 (22.8%)

DS: Down syndrome; FPIES: Food Protein Induced Enterocolitis Syndrome.

**Table 2 jcm-11-04047-t002:** Clinical characteristics of DS vs. non-DS patients with cow’s milk FPIES.

No. of FPIES Patients	40	
	*DS Population 9*	*Non-DS Population 31*
Sex		
Male (%)	5 (55.6%)	19 (61.3%)
Female (%)	4 (44.4%)	12 (38.7%)
Route of delivery (%)		
Vaginal delivery	4 (44.4%)	11 (35.5%)
Dystocic delivery	2 (22.2%)	3 (9.7%)
Cesarean section	3 (33.4%)	17 (54.8%)
Age (months) at first reaction		
Mean (SD)/median	1.65 (2.22)/0.5	3.13 (4.27)/1.75
Range	0–6	0–18
No. of episodes before diagnosis		
Mean (SD)/median	4.66 (3.67)/3	2.38 (1.08)/2
Range	1–13	1–5
Offending foods, *n* (%)	*41 acute reactions*	*101 acute reactions*
Milk	36 (87.9%)	96 (95.2%)
Egg		1 (0.9%)
Beef	1 (2.4%)	4 (3.9%)
Soy	4 (9.7%)	
Time (h) to the onset of symptoms		
Mean (SD)/median	1.55 (0.76)/1	2.27 (1.33)/2
Range	1–4	1–5
The severity of the acute FPIES reactions (%)	*41 acute reactions*	*101 acute reactions*
Mild-moderate	15 (36.6%)	43 (42.6%)
Severe	11 (26.8%)	31 (30.7%)
Very severe	15 (36.6%)	27 (26.7%)

DS: Down syndrome; FPIES: Food Protein Induced Enterocolitis Syndrome.

## Data Availability

Data is contained within the article.

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
