# Peer review of "Down Syndrome in FPIES: An Overwhelming and Unexpected Prevalence"

_jcm, 2022, doi:10.3390/jcm11144047_

Round 1

Reviewer 1 Report

This is a nice report on a possible increased risk for PFIES in patients with DS, which had not been appreciated before. It is well known that DS is associated with immune-related diseases and GI problems. The reported increased risk is very interesting and the authors were careful in showing that  cesarean section would not account significantly to the results.

Author Response

Dear Reviewer,
we really appreciated your comments. The case-series described shed new light on a disease characterized by various aspects that are still unclear.

Reviewer 2 Report

The authors review their practice's experience with food protein induced enterocolitis and described an increase incidence among children with Down Syndrome

1.     There is no need to put the number of females and males, with just the total number and one gender it is obvious that the rest will be of the other gender.

2.     The authors put in Material and Methods the Results. Line 69-71 and Table 1 belong to Results.

3.     Authors should not present in Results what is already shown in Table 1, for example number of patients with FPIES seen, their gender, etc.

4.     Although cesarean delivery may be associated with increased risk for allergy, in this case, the proportion of CS delivery in DS children was lower than that in the non-DS. In any case, the number of children with DS is so small that it would be difficult to draw any conclusions.  Moreover, this is not mentioned again in the Discussion.

5.     The authors make no mention of the potential bias that they may have in the patient referral pattern. Being a specific Allergy section in a tertiary care hospital in a capital city, doctors may be referring patients with underlying problems such as those in children with DS in a greater proportion that children without DS and, therefore, their unexpected findings.  This should be added to the Discussion.

6.     The number of references seems to be excessive for such a short paper.

Author Response

Dear Reviewer,
we have made the changes in the manuscript following your instructions. We hope we have met your expectations.
We are grateful for the suggestions provided which have improved the manuscript.

1.     We inserted only the number of males and the percentage. 
2.     We inserted the line 69-71 in the "Result" chapter.
3.     We have modified the manuscript according to the indications
4.     We have included in the manuscript a comment relating to the type of delivery
5.     In the discussion there is a comment on the possible influences related to the fact that the study was conducted in a reference center for pediatric diseases.
6.     We have reduced the number of references.

Reviewer 3 Report

This interesting and import, generally well-written report, contain several inappropriate word choices and would benefit from review by a native English speaker.

20-21 the word beef refers to cow meat so beef meat is redundant. Was fish the last identified food? What is the meaning of “even with”? Was fish significant? Recommend beef and fish be included with the list of other foods

26 recommend …This series provides a starting point…

The first paragraph of the introduction adds little to the subject of the report and should be shortened. Readers who want to know more about DS can find the information.

59 recommend … describe the cases of two

62 recommend suggested instead of supposed

Table I

Unless the mode of delivery is clearly related to FPIES, it should be deleted

The number of patients affected by each food should be listed

The species of fish, if know should be listed.

What fruit?

Delete 96-98

Tables I and II appear to be highly redundant and could be combined.

The age of DS didn’t drop from non-ds, it was just different.

The earlier onset in DS patients in general was not confirmed by the age of milk FPIES patients it is merely in the finding of a subset of patients.

159 caseload is the wrong word. Recommend record review instead

This is a retrospective record review rather than a cohort study

Author Response

Dear Reviewer,

we have made most of the suggested changes to the manuscript. 

Regarding the type of delivery, we wanted to include it in the manuscript because we believe that its influence on the composition of the microbiota may play a role in the onset of the syndrome.

We preferred to list the number of reactions for each food because different subjects of our series reacted to multiple foods and this could have confused the readers.

Regarding the tables, we believe that merging the information into a single table may cause confusion for readers. We fully realize that the two tables can be redundant, but we wanted to highlight the characteristics of the subgroup of acute cow's milk FPIES to emphasize subjetcs affected by DS.

The age at first reaction of patients with DS is lower because FPIES to cow's milk prevails in this subgroup and typically starts in the first weeks of life. 

We hope we have met your expectations.
We are grateful for the suggestions provided which have improved the manuscript.

This manuscript is a resubmission of an earlier submission. The following is a list of the peer review reports and author responses from that submission.